# Segmentation of Cerebral Arteries in 4D CT Perfusion: Effects of Temporal Data Augmentation and Skull Stripping

**Alena-Kathrin Golla**[1]                          Alena-Kathrin.Golla@medma.uni-heidelberg.de
**Lara-Jasmin Behrend**[1]                          Lara.Behrend@medma.uni-heidelberg.de
[1] *Computer Assisted Clinical Medicine, Mannheim Institute for Intelligent Systems in Medicine,*
*Medical Faculty Mannheim, Heidelberg University, Germany*

**Andreas Ziebart**[2]                          Andreas.Ziebart@umm.de
[2] *Department of Neurosurgery, University Medical Centre Mannheim, Heidelberg University, Germany*

**Anish Raj**[1]                          Anish.Raj@medma.uni-heidelberg.de
**Lothar R. Schad**[1]                          Lothar.Schad@medma.uni-heidelberg.de
**Frank G. Zöllner**[1]                          Frank.Zoellner@medma.uni-heidelberg.de
**Sherif A. Mohamed**[3]                          Sherif.mohamed@med.uni-heidelberg.de
[3] *Departement of Diagnostic and Interventional Radiology, University Hospital Heidelberg, Heidelberg University, Germany*

**Editors:** Under Review for MIDL 2021

## Abstract

Computed tomography perfusion (CTP) is used to quantitively evaluate the hemodynamics of the brain parenchyma in the assessment of cerebrovascular diseases like stroke or vasospams. We explore the segmentation of cerebral arteries from dynamic CTP on an initial data set of 20 patients. We use a 3D U-Net in combination with ratio-based sampling to extract the arterial vessel tree from 4D data. We investigate the benefit of temporal data augmentation and skull stripping. A shift of time frames by up to three time points in combination with skull stripping increased the DSC by 19.3% and decreased the ASSD by 38.2%.

**Keywords:** Vessel Segmentation, Computed Tomography Perfusion, Data Augmentation.

## 1. Introduction

Cerebrovascular diseases are the second leading cause of death after heart disease worldwide and the third leading cause of disability (Johnson et al., 2016). During an acute stroke, computed tomography (CT) is used for diagnosis. To provide more accurate information about the affected blood vessels, simple native CT is supplemented with with CT-angiography and CT-perfusion (CTA & CTP) (Wintermark, 2005; Wintermark et al., 2009). CTA is considered the standard procedure for vascular analysis of the patient (Mendrik et al., 2010). On the other hand, 4D CTP images provide dynamic relationship between arterial, tissue and venous enhancement, whereas CTA depicts only the arterial phase. Segmentation of arterial vessel tree from CTP would make an additional CTA examination redundant.

Previous approaches for the segmentation of the cerebral vasculature have been proposed mainly for CTA (Ni et al., 2020) and magnetic resonance angiography (MRA) (Livne et al., 2019). Meijs et al. employed a 3D-Dense-U-Net for vein and artery segmentation in CTP,

but used a mean and a standard deviation image across the time domain as network input (Meijs et al., 2020).

We investigate the segmentation of the major cerebral arteries directly from 4D CT perfusion data. We examine the shifting of time frames as a possible data augmentation technique. Lastly, we test the combination of temporal data augmentation with skull stripping.

## 2. Methods

We retrospectively collected 4D CTP data from 20 patients suspected of having vasospasm. The data was selected, reviewed, and annotated by two medical professionals - a neurosurgeon and a neuroradiologist. Only proximal arteries of the anterior and posterior circulation were annotated. This corresponds to locations in which stenosis or occlusion potentially leads to serious neurological deficits. We use the training framework based on ratio-based sampling (Golla et al., 2020) that we have previously presented for abdominal vessel segmentation. It was extended for processing of 4D data. For the network we chose a 3D U-Net (Ronneberger et al., 2015; Çiçek et al., 2016) based architecture that is shown in Fig. 1. Each CTP scan in our data set consists of 27 time steps, which we pass as channels to the network. Training is performed using the Adam optimizer. We combine the Dice loss and the cross entropy loss.

An initial Baseline training without temporal data augmentation or skull stripping was performed. The networks showed low performance in cases where the bolus time point differed from the data set avarage. We additionally noticed false positive segmentations of bone. To make the networks more robust against bolus time point differences, we test, whether data augmentation via a shift of the time frames by up to $\pm 3$ (TS=3) or up to $\pm 5$ (TS=5) frames could lead to more robust segmentation. The total number of time frames is kept constant by removing excess frames and replicating end frames. We further test whether combining temporal data augmentation with skull stripping (SS) (Najm et al., 2019) can further increase segmentation results and remove the false positives. For our experiments we employ 5-fold-cross-validation. To evaluate the performance we use the Dice similarity coefficient (DSC), connectivity (C), Hausdorff distance (HD) and the average symmetric surface distance (ASSD).

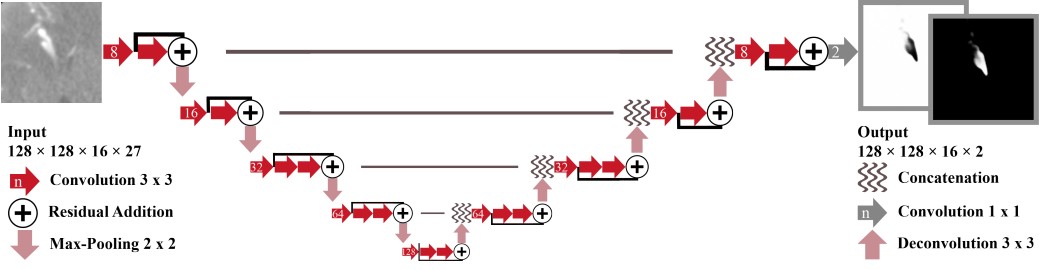

Figure 1: Network architecture

## 3. Results

The evaluation metrics for the different training configurations are summarized in Table 1. Segmentations from all networks for an examples case are shown in Fig. 2. Time frame shifting with both parameters improved the segmentation of the circle of Willis as seen in DSC and ASSD. However, C decreased slightly for TS=3 and stronger for TS=5. TS=3 lead to an increase in HD, while TS=5 performed similar to the baseline. Due to the better segmentation of the circle of Willis, we combined TS=3 with SS. TS=3 + SS achieved the best performance in our experiments for DSC, C and HD. This network barely produced false positive fragments, but also did not always capture all major arteries.

Table 1: Quantative evaluation - ↑ marks metrics for which higher values are better. ↓ marks metrics for which lower values are better.

| U-Net | DSC ↑ | C ↑ | HD ↓ | ASSD ↓ |
|-------|-------|-----|------|--------|
| Baseline | 0.394±0.238 | 0.434±0.214 | 78.956±19.836 | 17.263±16.244 |
| TS=3 | 0.456±0.191 | 0.428±0.237 | 104.258±32.974 | 12.351±10.257 |
| TS=5 | 0.450±0.136 | 0.380±0.203 | 78.290±26.366 | 10.318±6.690 |
| TS=3 + SS | 0.470±0.154 | 0.791±0.231 | 36.061±10.306 | 10.666±7.204 |

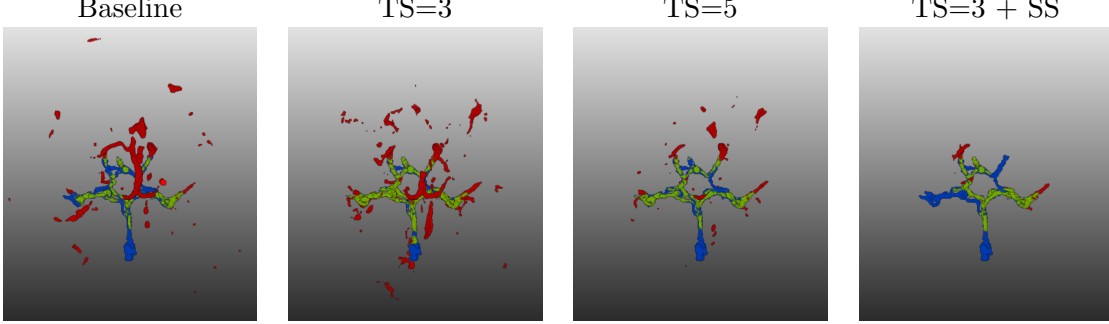

Figure 2: Segmentation results for an example case. True positive vessel segmentations are marked green, false negatives are marked blue and false positives are marked red, respectively.

## 4. Conclusion

Our results demonstrate that the introduction of the temporal data augmentation and skull stripping enhances segmentation quality. A combination of both methods yielded an increase of the DSC by 19.3% and a decrease of the ASSD by 38.2%. The experiments were performed on a limited data set of only 20 cases, but the results can be used to train segmentation networks on an extended data set in the future.

## Acknowledgments

This research project is part of the Research Campus $M^2OLIE$ and funded by the German Federal Ministry of Education and Research (BMBF) within the Framework "Forschungscampus: public-private partnership for Innovations" under the funding code 13GW0388A. We gratefully acknowledge the support of NVIDIA Corporation with the donation of the NVIDIA Quadro P5000 used for this research.

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
