# OpenReview forum: "Segmentation of Cerebral Arteries in 4D CT Perfusion: Effects of Temporal Data Augmentation and Skullstripping"
_MIDL.io/2021/Conference/Short — Submitted to MIDL 2021_

### Official Review · Reviewer_B5nv · 2021-04-29

**Confidence:** 3
**Final Rating:** 2

**Summary:**

The paper describes a method for segmentation of the major cerebral arteries in 4D CT perfusion images. The method is based on a 3D U-net. The paper proposes to feed 4D images to the network by placing the individual 3D volumes into the different channels of the input. This strategy is combined with a data augmentation strategy where the 4D data is shifted along the time-axis so that the sequence of 3D volumes is not always the same for an image. Furthermore, the paper investigates the use of a skull stripping algorithm to reduce the number of false-positives.

**Strengths:**

Using the 4D information for segmentation in 4D perfusion data makes a lot of sense. The paper is also clearly written and easy to follow. The experiments include a comparison with a baseline and report next to volume overlap measures also surface distance measures.

**Weaknesses:**

The paper investigates two concepts, namely 4D input for 3D segmentation and skull stripping as a pre-processing step. These are both rather obvious approaches and not very interesting from a methodological point of view. While it is nice to see their effects on the segmentation performance, the results are actually not very good, e.g., the Dice score is below 50% and the skull stripping apparently removes a lot of true-positives as well.

**Deanonymize Review:**

no

**Justification Of The Rating:**

See "Weaknesses" above. Additionally, neither data nor code are made available, the latter with the remark that the project is still in an early phase - maybe a bit too early even to publish method and results as these seem very preliminary.

**Paper Type:**

methodological development

**Special Issue:**

no

---

### Official Review · Reviewer_AS3c · 2021-04-30

**Confidence:** 4
**Final Rating:** 3

**Summary:**

The authors investigate the benefit of temporal data augmentation and skull stripping to segment cerebral arteries from 4D CTP data using a 3D U-Net. The performance of the network is evaluated using commonly used segmentation metrics. The results indicate that the proposed method combined with temporal data augmentation and skull stripping improves the segmentation results compared to not using any.

**Strengths:**

1.	The paper is concise and easy to understand.
2.	The overall motivation of the study is clearly described.
3.	Using 4D CTP data for such a task is an interesting research topic, especially for its application in clinical practice.
4.	Experiments and evaluation metrics are well thought. Thus, the results are convincing.


**Weaknesses:**

1.	Authors promote their work as being a methodological development and validation/application paper. However, there is no novel methodological contribution, as they use a conventional 3D U-Net, and shifting the time frames as a method for data augmentation has been previously explored. Nonetheless, its application for the current task is relevant, and therefore I think this paper is valuable.

2.	A small dataset is used to evaluate the network. Even though authors recognize this as a limitation, using a larger dataset is encouraged.

3.	More details about the dataset should be provided. Where was the data acquired? What is the image resolution? Looking at the input image in *Fig 1*, I assume that only a small patch containing the circle of Willis was used to train the network, but how were these patches selected?

4.	Authors analyze the data as 2D + time, but wouldn't the results be better if analyzing the data as 3D + time? I do not see any convincing motivation for using 3D data over 4D.


**Deanonymize Review:**

no

**Detailed Comments:**

1.	I would suggest doing a paired *t*-test to compare the results between the baseline and proposed experiments.

2.	Considering the ratio of TPs and TNs, it would be good to do an uncertainty assessment in the future.

3.	The paper could benefit from proofreading. I spotted a couple of typos:
- *Abstract ->* quantititively, vasospams
- *Table 1 ->* quantative
- *Methods ->* avarage
- *Introduction ->* double "with" in the third sentence


**Justification Of The Rating:**

Overall, the problem the paper tackles is relevant, and the proposed network combined with the proposed data augmentation and skull stripping is effective. Nevertheless, there is no technical novelty in this case. If this was only an application paper type, I would have given a strong accept.

**Paper Type:**

validation/application paper

**Special Issue:**

no

---

### Meta-Review · Program_Chairs · 2021-05-10

**Recommendation:** Reject
**Confidence:** 4

**Metareview:**

Though the topic is important and the approach appropriate, absence of methodological novelty combined with limited validation and mediocre results put this paper below the bar. The paper is also much longer than the allowed 3 pages.

---

### Decision · Program_Chairs · 2021-05-11

Reject